# A kinetically controlled platform for ligand-oligonucleotide transduction

Qiu-Long Zhang[1], Liang-Liang Wang[1], Yan Liu[1], Jiao Lin[1] & Liang Xu [1✉]

Ligand-oligonucleotide transduction provides the critical pathway to integrate non-nucleic acid molecules into nucleic acid circuits and nanomachines for a variety of strand-displacement related applications. Herein, a general platform is constructed to convert the signals of ligands into desired oligonucleotides through a precise kinetic control. In this design, the ligand-aptamer binding sequence with an engineered duplex stem is introduced between the toehold and displacement domains of the invading strand to regulate the strand-displacement reaction. Employing this platform, we achieve efficient transduction of both small molecules and proteins orthogonally, and more importantly, establish logical and cascading operations between different ligands for versatile transduction. Besides, this platform is capable of being directly coupled with the signal amplification systems to further enhance the transduction performance. This kinetically controlled platform presents unique features with designing simplicity and flexibility, expandable complexity and system compatibility, which may pave a broad road towards nucleic acid-based developments of sophisticated transduction networks.

[1] MOE Key Laboratory of Bioinorganic and Synthetic Chemistry, School of Chemistry, Sun Yat-Sen University, Guangzhou, China. ✉email: xuliang33@mail.sysu.edu.cn

Transduction of ligand signals, including small molecules and proteins, is highly essential and prevalent in both biomedical analysis and cellular regulations. For an analytical purpose, biological markers generally require a simple and efficient transduction strategy into detectable signals instead of direct measurements of these target molecules; in a typical operation of synthetic biology, specific ligands can bind to regulatory elements and induce signal transduction into the change of gene expressions. Compared with other signal transduction approaches, the ligand-oligonucleotide transduction is of particular uniqueness, as it provides the critical pathway to integrate non-nucleic acid molecules into all kinds of hybridization-based strand-displacement related processes[1,2]. Taking advantage of highly programmable and switchable base pairing, a variety of nucleic acid circuits and nanomachines have been constructed from test tubes to living cells based on strand-displacement reactions to perform kinds of tasks, such as acting as logic gates, making decisions, and transducing and amplifying signals[3–9], which have been widely applied in biomedical sensors and analysis[10–12], diagnostic and therapeutical nanomaterials[13–15], information technology[16–18], and synthetic biology[19–25]. However, in these nucleic acid-based developments, other molecules cannot be directly employed but require a "convertor" to achieve the ligand-oligonucleotide transduction[26–28]. Nucleic acid structures are highly dynamic and can be regulated by a variety of interacting ligands, including both small molecules and proteins. In some cases, ligand-binding can induce a conformational switch between different functional states of nucleic acids[28–31]. This is of particular significance as structural changes of nucleic acids may impact the strand binding and disassociation, which can be potentially employed in strand-displacement reactions[15,32,33]. The interactions between the ligands and nucleic acids provide the theoretical foundation to achieve the ligand-oligonucleotide transduction, but how to design a simple, general and effective converter is critically important.

The nucleic acid aptamer builds up an important bridge between nucleic acids and non-nucleic acid molecules[34–37]. Although kinds of aptamer-based designs have been developed to achieve signal transduction of ligands, such as electrochemical and fluorescent signals[38–41], not all of them can be involved with the strand-displacement reactions. Until now, the most popular approach for the integration of ligands into the strand-displacement reactions is based on the target-regulated strand competition, in which the ligand must compete with a partially complementary strand to bind on the aptamer. Different names have been adopted to describe this strategy, such as structure-switching aptamer[30,42–49], target-induced strand displacement or release[50–55], and duplexed aptamer[28], but the general principle of this approach relies on the utilization of a complementary strand (either a single-stranded oligonucleotide or a segment of engineered aptamer sequence) to partially mask the aptamer binding region. Once the ligand competes with the complementary strand and binds to the aptamer, disassociation of the pre-formed aptamer duplex would release the complementary strand or expose the toehold for strand invasion, which can be transduced into a detectable signal for the following process. In fact, in addition to these artificial developments, nature also employs a similar strategy to transduce the ligand signals into nucleic acids. For instance, riboswitch, a typical example that has been widely discovered in cells, is a class of regulatory RNA aptamers in which ligand-induced strand displacement of nucleic acids can regulate specific gene expressions[29,56,57]. Not limited to aptamers, some other types of ligand-nucleic acid interactions can also function through the competition behavior to regulate the strand dynamics[58]. Overall, taking this strategy of transducing the ligands into strand-displacement reactions, plenty of biosensors and biomaterials, as well as the gene-regulating applications of synthetic riboswitches, have been greatly developed.

Despite wide applications of competition-based transduction, this design still encounters some inevitable limitations. First, the sequence design of the complementary strand against the aptamer requires careful optimizations. A highly stable base pairing with the aptamer sequence would impede the competition performance, whereas an unstable duplex would induce unwanted background noise, resulting in the difficulty to control the balance between the transducing efficiency and the signal leakage. Second, to achieve at least partially masking the aptamer region, the complementary strand must be sequence-dependent on the aptamer strand, and consequently, it lacks the flexibility of sequence design. Since the competition-based transduction is highly dependent on the specific aptamer sequence, this would lead to the third point of limitations that logical operations between different ligands and aptamers are difficult to be directly modulated from the competition-based switches. In some reported studies, additional nucleic acid designs and circuits are needed to remove this sequence constraint[52,59]. The logical relationship between different ligands is highly important for both biomedical analysis and synthetic biology. Taking a potential application in synthetic biology as an example, if a high level of ligand A together with a low level of ligand B can be recognized as an effective indication for regulation of the target gene expression, one can envision that a direct check of the logical relationship between A and B (the "A not B" gate) would determine whether the following process could be triggered. Hence, there is a strong urge to generate a general and easy-to-design approach with great flexibility and controllability to achieve ligand-oligonucleotide transduction.

In this work, a general platform is constructed to transduce the signal of ligand into the release of independent oligonucleotide through precise kinetic control. This platform requires only a minimal strand design and can be employed to induce the transduction of both small molecules and proteins into desired sequences of oligonucleotides. More importantly, orthogonal, logical, cascading, and amplifying operations between different ligands can be efficiently established to generate all kinds of transduction systems. This platform, overcoming the limitations encountered in the competition-based transduction by great designing simplicity and flexibility, will pave a broad road for applications of ligand-oligonucleotide transduction in biomedical analysis, nucleic acid nanomaterials, nucleic acid computation, and synthetic biology.

## Results

**Designing principle for kinetic control of strand displacement.**
During the binding process between the ligand and the aptamer, the single-stranded oligonucleotide generally undergoes a dynamic conformational change from an unconstrained strand to a tightly stable structure. Even in the absence of any defined structural information, this characteristic feature of ligand-aptamer interaction can be directly employed as a regulatory element to control nucleic acid structures. In this scenario, a short duplex stem is introduced into the aptamer sequence besides the core binding domain (Fig. 1a). In the absence of the target ligand, the short duplex stem cannot maintain a stable hairpin structure due to the unconstrained binding domain, and therefore the aptamer sequence stays in an unstructured state. Only when the binding domain tightly interacts with the ligand, the short duplex stem would be stably formed due to restrained conformational dynamics of the aptamer sequence. From this perspective, the short duplex stem is regulated by the dynamic conformation of the aptamer binding domain; once the aptamer conformation is

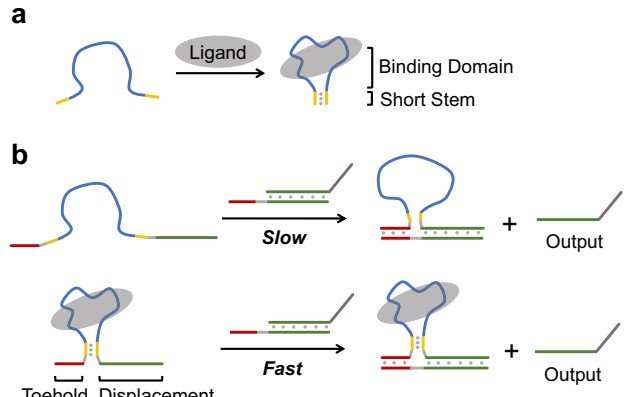

**Fig. 1 Kinetic control of ligand-oligonucleotide transduction. a** Structural alteration of the nucleic acid aptamer with a designed short stem. In the absence of the target ligand, the aptamer cannot form a defined structure, leading to the inability of the stem duplex. When the ligand binds to the aptamer, the stem duplex can be stably formed with assistance from the structured binding domain. **b** Kinetic distinction between the free and the ligand-bound invading strand. The aptamer sequence with the short stem is inserted between the toehold and displacement domains of the invading strand. In the free state, the toehold and displacement domains cannot be closely connected to function as an effective invading strand, resulting in a slow displacement reaction. When the ligand binds to the aptamer, these two domains are tightly linked together by the formation of the stem duplex, leading to a fast kinetic behavior of the strand-displacement reaction.

defined by ligand binding, the thermostability of the short duplex stem would be enhanced.

In a conventional toehold-mediated strand displacement, the invading strand can be divided into two segments: the toehold domain and the displacement domain. Previous studies on strand-displacement reactions indicated that the toehold and displacement domains can be separate and independent with either a remote or a combinatory control[60,61]. Notably, different designs of toeholds would significantly alter the kinetics of strand displacement. Hence, we envisioned that whether the aptamer with a short duplex stem could be introduced between the toehold and displacement domains to regulate the strand invasion. As described in Fig. 1b, in the absence of a binding ligand, the unstructured aptamer sequence separates the toehold from the displacement domain in a disorderly manner. Therefore, the subsequent strand displacement would be unable to proceed efficiently. If the short duplex stem is stably formed upon the ligand binding, the toehold and displacement domains would be tightly connected, leading to fast kinetics for strand invasion. With the control of strand displacement, the ligand signal can be governed in a kinetic approach to generate the output strand. In this platform, the efficiency of signal transduction is generally relied on how fast the strand displacement could proceed upon the ligand binding. Moreover, given that the strand-displacement reaction is independent of the aptamer sequence, the output strand can be flexibly designed with the minimal requirement to avoid strong and stable secondary structures between the aptamer sequence and the invading strand, which provides great flexibility to design all kinds of output strands for different types of nucleic acid circuits.

To verify the general principle for our concept of kinetic control, we first tested the kinetic difference between an unstructured and a hairpin-shaped DNA invading strand (Supplementary Fig. 1). In this design, the toehold and displacement domains were separated by either a free single-

stranded loop or a stable hairpin structure (a 25-nt length for both sequences). The reason we chose a 25-nt length as the spacer between the toehold and the displacement domain was that the length of an aptamer sequence is typically similar or more than 25-nt. A short sequence of DNA strand is, typically, less dynamic and more entropically confined than a long strand in terms of the conformational variation. If a short sequence can be kinetically controlled effectively, a longer length with more intensely conformational change would be manipulated more easily. In addition to the loop or hairpin sequence, a couple of extra conjugated nucleotides were also introduced into the invading strand to facilitate the strand-displacement reaction as reported previously[61]. As observed in Supplementary Fig. 1, the invading strand with an unstructured loop was much less efficient than the hairpin-structured strand based on the monitoring of the fluorescence reporter system, suggesting a great kinetic distinction between these two structural states. Notably, the length of the toehold impacted the kinetic distinctions. Increasing the length of toehold binding would raise the kinetic rate of the hairpin strand but bring along with the unwanted enhancement of the unstructured strand, whereas a shorter toehold would decrease the efficiency of the hairpin strand but greatly suppress the signal of the unstructured strand. Therefore, adjusting the length of the toehold provided a tunable pathway to regulate the kinetic performance between these two structural states. Nevertheless, this experiment explicitly demonstrated that the structural change of the spacer sequence between the toehold and the displacement domain could be employed to govern the kinetics of the strand displacement.

**Kinetically controlled transduction of small molecules.** Based on the concept described in Fig. 1, we then selected a widely investigated ATP DNA aptamer[62] to test the performance of our design. As depicted in Fig. 2a, the core binding domain along with a short duplex stem was introduced between the toehold and the displacement domain. The strand-displacement reaction was reported by the increasing fluorescence (Rep-1) as an indication for the output oligonucleotide. Herein, we chose a middle length (9 nt) of the toehold domain to initiate the strand invasion, and the length of the short stem is critical. We started the optimization of the short stem from a 4-bp duplex to a single base pair. As observed in Fig. 2b, the existence of 1 mM ATP greatly enhanced the efficiency of the strand-displacement reaction. Interestingly, the invading strand with a 4-bp duplex stem promoted the most efficient fluorescence increase when bound with ATP, but a slight leakage of the background signal in the absence of ATP was also observed. Weakening the stability of the duplex stem significantly reduced the background leakage in the absence of ATP along with a compromise of the strand-displacement efficiency in the presence of ATP. When the stem was reduced to a single base pair, the duplex could not be stably formed even in the presence of ATP, resulting in an inefficient displacement performance as described in the last panel of Fig. 2b. These results clearly suggested that the stability of the duplex stem could directly control the kinetic difference between the presence and the absence of the binding ligand. Overall, the remarkable distinction regulated by the binding of the ATP ligand indicated an effective and convenient approach to transduce the ligand signal into the release of the oligonucleotide.

From the optimizations in Fig. 2b, we selected a 3-bp stem (5'-CTT-3'), which presented a highly efficient displacement performance in the presence of ATP and a minimal background leakage, as a representative to check the sensitivity of this kinetic behavior when responding to the change of ATP concentration. As depicted in Fig. 2c, the kinetic performance of strand displacement was positively dependent on the variations of

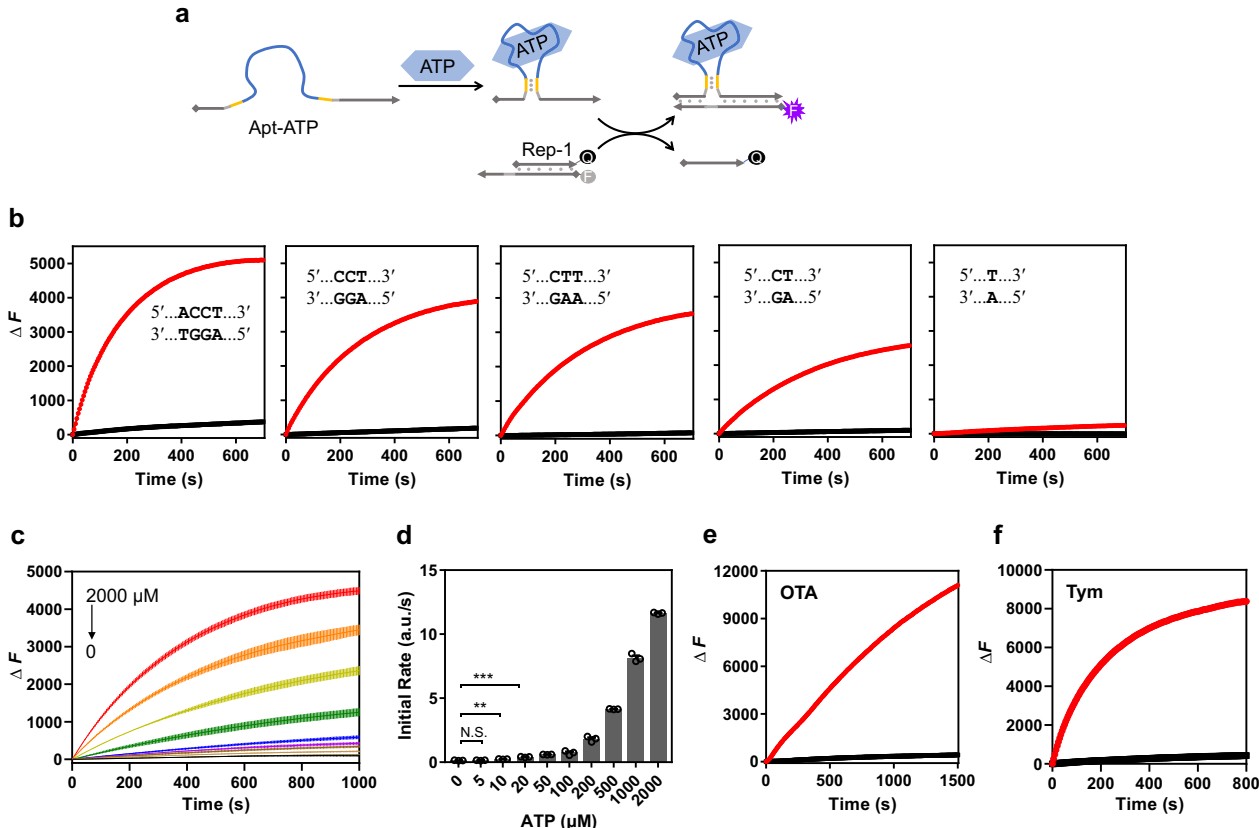

**Fig. 2 Kinetic control of ligand-oligonucleotide transduction regulated by small molecules. a** Design of the ATP-induced strand-displacement reaction. The ATP aptamer with a designed short stem was placed between the toehold and displacement domains of the invading strand (Apt-ATP). The addition of ATP promoted the strand displacement with Rep-1. The quenched fluorescence was then recovered after the strand displacement as an indication of the invasion process. Squares and arrows drawn on DNA strands represent 5′ termini and 3′ termini, respectively. Base-pairing is shown by gray dots. **b** Optimization of the stem duplex of Apt-ATP. The red curve indicated the kinetic data of the reporter system in the presence of 1 mM ATP. The black curve indicated the background signal without ATP. Different designs of the stem duplexes were labeled in the corresponding panels. **c** Concentration-dependent kinetic performance of the ATP-induced strand-displacement reaction. The design of the stem duplex was 5′-CTT-3′. The concentrations of ATP were 2000, 1000, 500, 200, 100, 50, 20, 10, 5 and 0 μM, respectively. **d** Analysis of initial kinetic rates for the change of fluorescence in the presence of different concentrations of ATP. "a.u.": arbitrary units for fluorescence. Data in **c**, **d** are presented as mean values with standard deviations (error bars) derived from three independent experiments. "N.S.": no significant difference; **$P = 0.0038$, ***$P = 0.0003$ (unpaired $t$ test, two-tailed $P$ value, $n = 3$). **e**, **f** Representative kinetic performance of the mycotoxin ochratoxin A (OTA) and the ʟ-tyrosinamide (Tym) induced strand displacement-based the same design. The red curves indicated the kinetic data of the reporter system in the presence of 50 μM OTA or 100 μM Tym, respectively. The black curves indicated the background signal without ligands. Source data are provided as a Source Data file.

ATP concentrations. Quantitative analysis of initial rates of the fluorescence increase indicated when the ATP concentration was lowered to 10 μM, the kinetic difference was still clearly distinguished compared with the absence of ATP (Fig. 2d). These initial rates could be fitted by a linear line with target ATP concentrations below 1 mM (Supplementary Fig. 2), and a limit of detection (LOD) was determined to be 7.6 μM based on a $3\sigma_b/$ slope, where $\sigma_b$ is the standard deviation of three blank samples. In fact, not limited to the defined simple solution, this platform could also function in the complex system for ATP detection as observed from the recovery test in the 10% serum samples (Supplementary Fig. 3). These results suggested that the kinetic behavior of our design was effectively controlled by the ligand binding and could directly reflect the concentration change of the target with a good sensing performance.

Moreover, in addition to the ATP binding aptamer, we also examined another two ligand-binding aptamers, the mycotoxin ochratoxin A (OTA)[63–65] and ʟ-tyrosinamide (Tym)[66] DNA aptamers. Until now, no structural information regarding these two aptamer sequences was reported. Therefore, we needed to

design the aptamer-based strand displacement according to our strategy without consideration of any possible folding conformation. Herein, an extra short duplex stem was added with each aptamer sequence. Notably, in order to allow the free space for the aptamer folding and minimize possible impacts on the aptamer binding caused by the formation of the duplex stem, a couple of flanking nucleotides between the aptamer binding domain and the short duplex stem were introduced. For the OTA aptamer, with optimization for the length of the short duplex stem from 4-bp to 2-bp (Supplementary Fig. 4), we identified the consensus binding sequence with a 2-bp stem that could achieve an efficient response against OTA with a minimal background signal (Fig. 2e). Quantitative analysis indicated the kinetic rates were fully dependent on the concentrations of OTA, which could also be utilized as a sensing strategy (Supplementary Fig. 5). Similarly, through the addition of an extra duplex stem with the Tym aptamer, the strand displacement was also kinetically controlled by the Tym ligand in a concentration-dependent manner (Fig. 2f, Supplementary Figs. 6 and 7). These results indicated that even in the absence of structural information, the

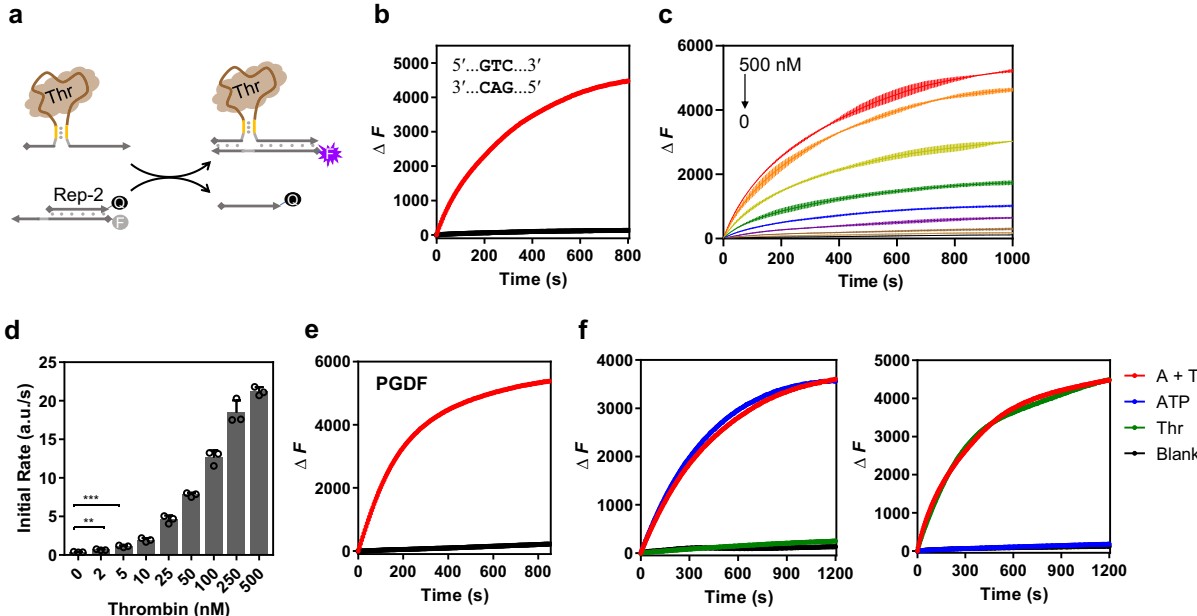

**Fig. 3 Protein-regulated kinetic control of ligand-oligonucleotide transduction. a** Design of the thrombin (Thr) induced strand displacement reaction. Thrombin promoted the strand displacement with Rep-2 with increased fluorescence signals. **b** Representative kinetic performance of thrombin-induced strand displacement reaction. The design of the stem duplex was 5′-GTC-3′. The red curve indicated the kinetic data of the reporter system in the presence of 200 nM thrombin. The black curve indicated the background signal without thrombin. **c** Concentration-dependent kinetic performance of the thrombin-induced strand-displacement reaction. The concentrations of thrombin were 500, 250, 100, 50, 25, 10, 5, 2, and 0 nM, respectively. **d** Analysis of initial kinetic rates for the change of fluorescence in the presence of different concentrations of thrombin. "a.u.": arbitrary units for fluorescence. Data in **c**, **d** are presented as mean values with standard deviations (error bars) derived from three independent experiments. **P = 0.0065, ***P = 0.0006 (unpaired t test, two-tailed P value, n = 3). **e** Representative kinetic performance of the PDGF protein-induced strand-displacement reaction-based the same design. The red curve indicated the kinetic data of the reporter system in the presence of 50 nM PDGF. The black curve indicated the background signal without PDGF. **f** Orthogonal manipulation between ATP and thrombin. Independent transduction platforms for ATP and thrombin co-existed in the same system. The ATP transduction was designed to be reported by Rep-1; the thrombin transduction was designed to be reported by Rep-2. Rep-1 and Rep-2 were fully independent without any sequence correlation. The signals of Rep-1 (left panel) and Rep-2 (right panel) were the fluorescence of the FAM and TMR fluorophores, respectively. The kinetic data in presence of both 1 mM ATP and 200 nM thrombin (the A+T curve), only 1 mM ATP (the ATP curve), and only 200 nM thrombin (the Thr curve) were measured and compared along with the background signal (the Blank curve). Source data are provided as a Source Data file.

ligand-aptamer interaction could still be utilized in our design as a controlling factor to regulate the kinetics of strand displacement. Collectively, our systematical investigations demonstrated that our design of ligand convertor could achieve highly efficient and sensitive transduction from a small molecule to a signal of output oligonucleotide based on the kinetic control of the strand-displacement reaction.

**Protein-oligonucleotide transduction with orthogonality.** Since a small molecule can be integrated into the strand-displacement reaction, we further tested whether a large ligand, such as a protein, would be also introduced into our design. To this end, we first selected a well-studied protein, thrombin, as an example. The same designing strategy was adopted with its DNA aptamer[67] as described in Fig. 3a, and the measurements of fluorescence change (Rep-2) would reflect the process of the strand displacement. Similarly, we selected a 9-nt toehold and optimized the length of the short duplex stem. The length of the stem was varied from 4-bp to 0-bp with gradually decreasing the kinetic rates of the strand displacement (Supplementary Fig. 8). Among these different lengths of the stems, the aptamer design with a 3-bp duplex exhibited a high kinetic efficiency along with a weak background signal (Fig. 3b). These data revealed that even a protein-ligand could still be effectively transduced in our design. Moreover, the kinetic efficiency was directly dependent on the concentration of the thrombin ligand. Taking the 3-bp aptamer as

a representative, we observed that the fluorescence kinetics of the strand displacement was attenuated along with the concentration of thrombin reduced from 500 to 0 nM (Fig. 3c), demonstrating a positive dependency between the thrombin ligand and the output oligonucleotide. Quantitative analysis of initial rates of the fluorescence increase indicated even if the thrombin concentration was as low as 2 nM, the kinetic difference was still clearly distinguished compared with the blank sample (Fig. 3d). Fitting of these kinetic rates from 0 to 50 nM by a linear line indicated LOD of thrombin was 0.96 nM (Supplementary Fig. 9). Moreover, the recovery test in the 1% serum samples showed that this platform could also be utilized to detect thrombin in the complex system (Supplementary Fig. 10). These data together suggested a good protein-sensing performance based on this design. Besides the thrombin protein, we also studied another protein, platelet-derived growth factor (PDGF). The DNA aptamer against PDGF possessed a longer sequence than the thrombin aptamer[68]. Employing the same strategy, optimization of the short duplex stem suggested similar behaviors as the thrombin aptamer (Supplementary Fig. 11). The aptamer with a 2-bp duplex stem presented an excellent balance between the displacement efficiency and the background leakage (Fig. 3e). Concentration-dependent kinetic analysis on PDGF also revealed a sensitive and quantitative responding performance (Supplementary Fig. 12). Investigations of these protein ligands showed that our design of ligand-oligonucleotide transduction could function in general with both proteins and small molecules.

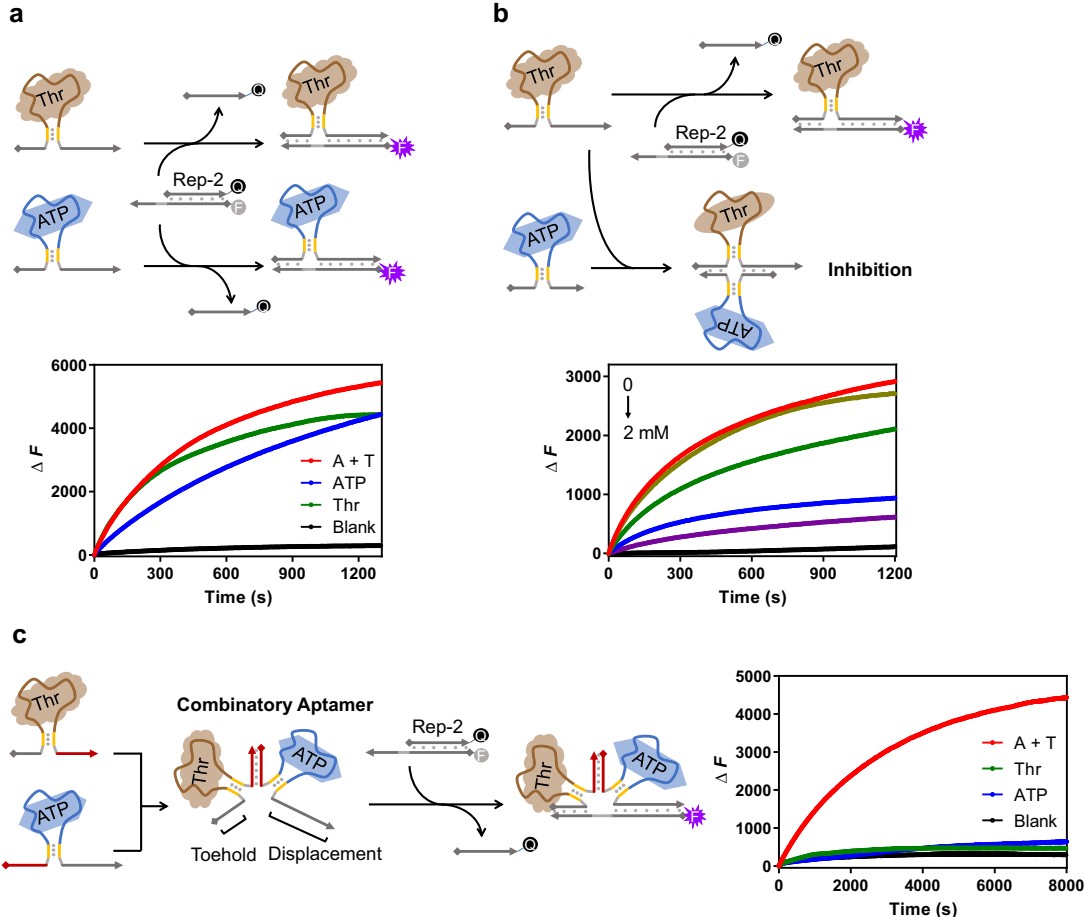

**Fig. 4 Logical operations between different ligands. a** The "OR" logic gate between ATP and thrombin. ATP and thrombin were designed to promote the strand-displacement of the same reporter system. The kinetic data in presence of both 0.5 mM ATP and 100 nM thrombin (the A+T curve), 1 mM ATP (the ATP curve), and 200 nM thrombin (the Thr curve) were measured and compared along with the background signal (the Blank curve). **b** The "NOT" logic gate between ATP and thrombin. The ATP aptamer was designed to function as an inhibitor against thrombin transduction. The concentration of thrombin was 200 nM; the concentrations of ATP were 0, 0.2, 0.5, 1, and 2 mM, respectively. The black curve indicated the background signal without thrombin or ATP. **c** Utilization of combinatory aptamer to build the "AND" logic gate between ATP and thrombin. The two aptamer designs were combined to function together through the joint duplex. The kinetic data in presence of both 1 mM ATP and 200 nM thrombin (the A+T curve), only 1 mM ATP (the ATP curve), and only 200 nM thrombin (the Thr curve) were measured and compared along with the background signal (the Blank curve). Signals of all these strand-displacement reactions were reported by the Rep-2 system. Source data are provided as a Source Data file.

Given that different ligands selectively bind to the corresponding aptamers, orthogonal manipulation among different ligand-oligonucleotide transduction systems could be achievable. To show the orthogonality of this design, we prepared two transduction platforms. One system functioned based on the ATP aptamer, which was reported by Rep-1 with the FAM fluorophore; the other one only worked in the presence of thrombin, which was reported by Rep-2 with the TMR fluorophore (Fig. 3f). These two platforms were mixed together into the same system. Once the ATP was added into the mixture, only the FAM fluorophore was reported with increased fluorescence, whereas the TMR fluorophore remained unaffected. In contrast, if the thrombin was mixed into the solution, the TMR fluorescence was increased along with the unaffected FAM fluorophore. Only in the presence of both ATP and thrombin, the fluorescence increase of the FAM and TMR fluorophores could be detected at the same time. This orthogonality demonstrated that different transduction platforms can co-exist in the same system without any crosstalk and the function of the platform was solely based on the target ligand, which further demonstrated the universality of this design.

**Transduction systems with logical operations**. As mentioned above, although the toehold and displacement domains were regulated by the ligand-aptamer interaction, their sequence design of the invading strand was independent of the aptamer. Therefore, the flexibility of sequence design provided great convenience to construct the logical operations between different ligands. We selected the ATP and thrombin ligands as the proof of principle, and generated "OR", "NOT", and "AND" gates as representatives (Fig. 4).

For the construction of the OR gate, the toehold and displacement domains shared the same sequence in both the ATP and thrombin transduction designs (Fig. 4a). Thus, either ATP or thrombin was capable of activating the strand displacement and consequently being detected by the reporter system. As observed in the kinetic curves in Fig. 4a, in the absence of ATP and thrombin, no significant fluorescence increase was detected; if either one of these two ligands was added into the system, the ligand-oligonucleotide transduction would be triggered and reported by the fluorescence increase, indicating a typical "OR" relationship between ATP and thrombin.

To construct a NOT relationship, the thrombin system was maintained as the regular transduction design, whereas the two single-stranded segments linked through the stem of the ATP aptamer were designed as complementary sequences against the toehold and the displacement domain of the thrombin system as depicted in Fig. 4b. In this design, ATP functioned as an inhibitor toward the thrombin system. In the absence of ATP, these two single-stranded segments in the ATP aptamer design are maintained in a free state, and it would be difficult to effectively mask the invading domains of the thrombin system. The addition of ATP would make them tightly connected through the short duplex stem and strengthen the base-pairing against the thrombin system, leading to the inhibited reporting process. These behaviors were directly revealed by the kinetic curves in Fig. 4b. With the increase of ATP concentrations, the transduction efficiency of thrombin was significantly reduced, explicitly showing a "thrombin not ATP" mode. Notably, interactions between the thrombin and ATP aptamer-containing strands were needed before the reporting process as the inhibition and the strand displacement were in direct competition.

To generate an AND gate, we introduced the concept of a "combinatory aptamer". In this design, the toehold domain was linked with the thrombin aptamer and the displacement domain was linked with the ATP aptamer. These two aptamer systems could be bound together through the complementary sequence to form a combinatory aptamer as described in Fig. 4c. In the absence of thrombin, the short duplex stem in the thrombin part of this combinatory aptamer would not be stable, and the toehold could not be connected together with the displacement domain. Similarly, in the absence of ATP, the displacement domain would not be tightly linked with the toehold. Hence, either the absence of thrombin or ATP would not initiate an efficient strand-displacement reaction as observed in the kinetic curves of Fig. 4c. Only in the presence of both thrombin and ATP, the combinatory aptamer could form a stable structure with the tightly linked toehold and displacement domains, which possessed the ability to trigger the strand displacement. In this combinatory aptamer, the length of the toehold was further optimized to ensure an efficient strand displacement (Supplementary Fig. 13). Through a combination of two aptamers, an "AND" was successfully constructed between ATP and thrombin. Overall, by direct manipulation of the transduction design, the logical relationship between different ligands can be established with great designing flexibility.

**Cascade relationship between different transduction systems.** Besides the logical operations between different ligands, the cascading transduction was also successfully constructed, in which one ligand was the initial trigger for the following ligand transduction. Herein, we developed two types of cascading systems with ATP and thrombin DNA aptamers as examples. In these two systems, thrombin was the prerequisite for ATP transduction. In the first system (described in Fig. 5a), the displacement domain of the invading strand for the ATP transduction was masked by the complementary strand. As a result, even in the presence of ATP, this aptamer structure could not effectively react with the reporter system. Only when thrombin initiated the strand release of the displacement domain for the ATP transduction, the following strand displacement could be observed as monitored by the increase of fluorescence (kinetic curves in Fig. 5a). In the other design, the indispensable toehold for the ATP transduction was masked by the complementary strand, and therefore, it required activation from the thrombin transduction to be released (Fig. 5b). Through masking either the displacement domain or

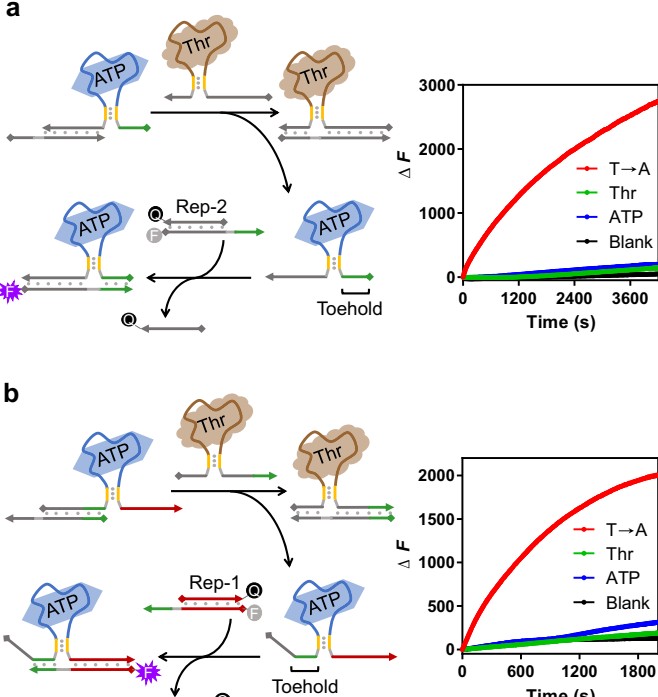

**Fig. 5 Cascade reactions between different ligands. a** Release of the displacement domain for the cascade reaction. The addition of thrombin triggered the release of the masked displacement domain for the ATP transduction, and subsequently, the reporter system was activated by the ATP-induced strand displacement. Signals of these strand-displacement reactions were reported by the Rep-2 system. **b** Release of the toehold domain for the cascade reaction. The addition of thrombin triggered the release of the masked toehold domain for the ATP transduction, and subsequently, the reporter system was activated by the ATP-induced strand displacement. Given that the toehold domain was generally too short to maintain a stable duplex at room temperature, an extended strand was utilized to stably inhibit the toehold. Signals of these strand-displacement reactions were reported by the Rep-1 system. The kinetic data in presence of both 1 mM ATP and 200 nM thrombin (the T→A curve), only 1 mM ATP (the ATP curve), and only 200 nM thrombin (the Thr curve) were measured and compared along with the background signal (the Blank curve). Source data are provided as a Source Data file.

the toehold, both designs exhibited a good efficiency to trigger the cascading transduction. With this concept of cascade transduction, we further generated two types of feedback circuits between the ATP and thrombin ligands. One was stimulating feedback with a positive effect, in which the signal of thrombin could be enhanced by the addition of ATP through the cascade transduction process, whereas ATP alone could not produce significant signals (Supplementary Fig. 14). The other one was the opposite mode with inhibiting feedback between thrombin and ATP, which was also relied on the cascade transduction design (Supplementary Fig. 15). The simplicity of our cascade design provided many possibilities to expand the complexity of the ligand-oligonucleotide transduction.

**Transduction coupled with other nucleic acid nanomachines.** Since an independent oligonucleotide could be released from the transduction of a ligand, theoretically, this output strand could be involved in any desired strand-displacement reactions. Herein, two kinds of signal amplification strategies were coupled after the ligand-oligonucleotide transduction to check the connectivity and

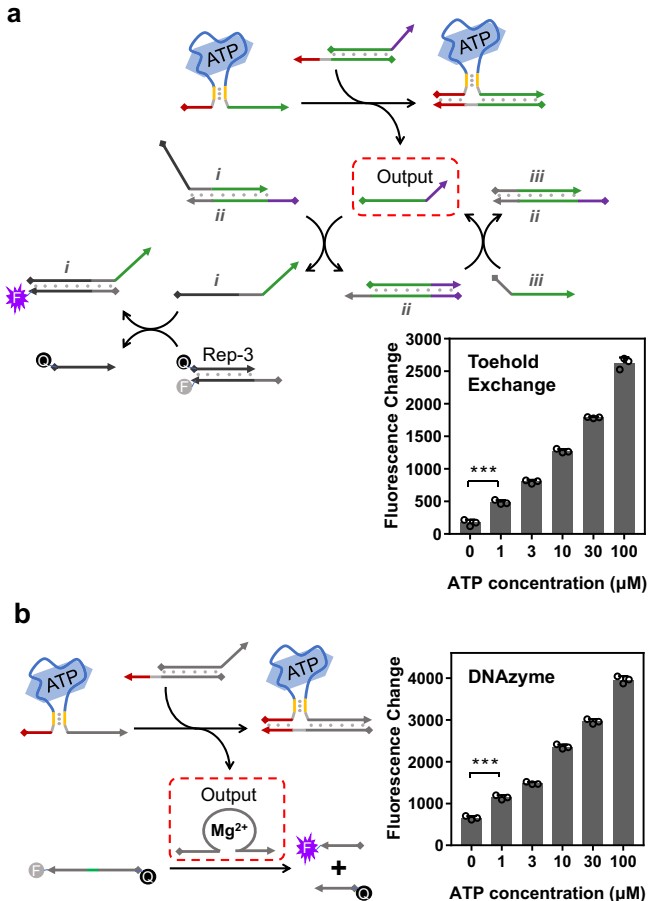

**Fig. 6 Ligand-oligonucleotide transduction coupled with other nucleic acid-based techniques. a** The ATP-induced oligonucleotide output coupled with the toehold exchange-based catalysis. Once the output strand was released upon the addition of ATP, the following toehold exchange system was initiated with amplified output signals. The fluorescence change after 4-h incubation in the presence of different concentrations of ATP was shown in the columns. Signals were reported by the Rep-3 system. ***$P$ = 0.0008 (unpaired $t$ test, two-tailed $P$ value, $n$ = 3). **b** The ATP-induced oligonucleotide output coupled with the magnesium-based DNAzyme system. The restrained DNAzyme strand was released upon the addition of ATP. The dual-labeled substrate was cleaved by the DNAzyme with increased fluorescence signals. The fluorescence change after 1-h incubation in the presence of different concentrations of ATP was shown in the columns. ***$P$ = 0.0003 (unpaired $t$ test, two-tailed $P$ value, $n$ = 3). Data in **a** and **b** are presented as mean values with standard deviations (error bars) derived from three independent experiments. Source data are provided as a Source Data file.

compatibility of this platform with other nucleic acid nanomachines. First, a toehold exchange-based catalysis systems[69], an extensively studied nucleic acid amplification technique, was introduced after the ligand transduction. As described in Fig. 6a, once the output strand was released from the ATP-induced strand displacement, the subsequent signal amplification was initiated. Fluorescence signals were significantly enhanced along with incubation of the catalysis system (Supplementary Fig. 16). After a 4-h amplification, even a 1 μM concentration of ATP could be significantly detected (columns in Fig. 6a). These results suggested that our ligand-oligonucleotide transduction platform could be effectively coupled with toehold exchange to generate amplified signals, which could achieve sensitive ligand detection. In addition, we also examined the connectivity of this platform

with the DNAzyme system (Fig. 6b). A magnesium-mediated cleavage DNAzyme[70] was generated after the ligand-oligonucleotide transduction. Initially, the DNAzyme sequence was partially masked by the complementary strand to abolish its enzymatic activity; the addition of ATP into the transduction system could release the inhibited DNAzyme. Consequently, the fluorescence enhancement was then observed when the fluorophore-labeled substrate was cleaved by the DNAzyme (Supplementary Fig. 17). Within 1 h of DNAzyme-based signal amplification, low concentrations of ATP could also be clearly monitored as shown in the columns of Fig. 6b. With the coupling of these two signal amplification techniques, LOD of ATP was calculated to be in the sub-micromolar level (Supplementary Fig. 18, ~0.93 μM in the toehold exchange system and ~0.75 μM in the DNAzyme system). Similarly, thrombin, as a representative for the protein–ligand, could also be integrated into the toehold exchange-based catalysis (Supplementary Fig. 19) and DNAzyme systems (Supplementary Fig. 20) to further enhance its sensitivity. Quantitative analysis revealed LOD of thrombin was lowered by ~3-fold with these two amplification processes (Supplementary Figs. 21 and 22). Taken together, these results demonstrated this ligand-oligonucleotide transduction platform could be directly coupled and integrated into other nuclei acid nanotechnologies. Notably, due to the flexibility of this platform, future designs with the employment of different signal sensing and amplifying techniques are fully foreseeable to greatly expand the application scenarios of our platform.

## Discussion

In summary, we constructed an efficient system that can achieve general transduction from ligands to any desired oligonucleotides. In this design, the binding sequence of aptamer with a short duplex stem was introduced between the toehold and displacement domains of an invading strand. Employing the structural alteration during the ligand–aptamer interaction, kinetic control of the strand-displacement reaction was precisely regulated, which in turn could transduce the ligand signal into the strand output. With this concept, we successfully applied our platform to achieve the transduction of five different ligands including both small molecules and proteins. More importantly, due to the designing convenience and flexibility, orthogonal transduction, logical operations, and cascading manipulation between different ligands were all established efficiently. Besides, this transduction system could be directly coupled with other nucleic acid nanomachines to further amplify the ligand transduction performance.

Our design, as a general platform for ligand-oligonucleotide transduction, presents a few superior features compared with previously reported strategies. (i) Designing simplicity. The aptamer sequence can be directly placed into an invading strand even in the absence of any structural information regarding the interactions between the aptamer and the ligand. Some aptamers can form a defined stem structure while some may not (e.g., the OTA and Tym aptamers investigated in this work). Through either simple optimization or extra addition of the stem duplex, a great balance between the transduction efficiency and the signal leakage can be easily achieved. (ii) Designing flexibility. Theoretically, the ligand can be transduced into any desired nucleic acid sequences, provided that no strong and stable secondary structures can be formed between the aptamer sequence and the invading strand. This is because the aptamer can be inserted into a random strand with great designing flexibility. As a result, the output oligonucleotide can be fully independent of the aptamer sequence. Therefore, orthogonal and multiple manipulations of ligand transduction can be easily achieved as well. (iii) Expandable complexity. In this work, we examined several distinct logical

operations and cascade reactions between two ligands. Obviously, one can envision multiple regulations among more ligands that could be also achievable to enrich the function of this platform. It is also necessary to mention that the logical relationship in this study was constructed from the invading strands. Since the output strand is fully unrelated to the aptamer sequence, establishing the logical operations from the output strand would be also foreseeable and potentially provide more versatile approaches to expand the complexity of this system. Notably, compared with the downstream operations, direct manipulation of the invading strands possesses one important advantage: ligand-induced interactions between aptamer-containing strands can be directly detected by the reporter system for signal transduction without any intermediate strand components. These simplified logic operations might be suitable to construct ligand-controlled synthetic circuits for gene manipulation in cellular contexts. (iv) System compatibility. Herein, we successfully connected the ligand-oligonucleotide transduction with the toehold-mediated strand exchange reaction and the DNAzyme catalysis to generate a high ligand sensitivity. Given that the output strand is freely designed, it is fully expectable that our design can be also coupled with other strand-displacement reactions, including non-enzymatic catalysis as well as enzymatic amplification, to provide diverse options for different requirements of ligand transduction[2,10]. This system compatibility will greatly enrich the applicable scenarios of this platform. Hence, all kinds of sophisticated transduction networks can be potentially established based on this platform.

While we utilized DNA strands to establish this transduction platform for strand-displacement reactions in this work, the RNA system would be expected to function similarly. Moreover, in addition to the fluorescence-based sensing investigated in this study, with this design of ligand transduction, we anticipate different kinds of small molecules and proteins would be easily involved in the strand-displacement reactions for various purposes. To name a few, the ligand may be utilized to control the formation and deformation of DNA nanostructures, to manipulate gene expressions, or to function as input for DNA computations. This kinetically controlled platform for ligand-oligonucleotide transduction paves a broad road towards these developments.

## Methods

**Materials**. All the chemicals used in our experiments were of analytical purity and purchased from Sangon Biotech. Recombinant human PDGF BB was purchased from Novoprotein. Thrombin was purchased from Macklin. Ochratoxin A (OTA) and L-tyrosinamide (Tym) were purchased from Aladdin. ATP and human serum were purchased from Sangon Biotech. All DNA sequences including fluorophore-labeled strands used in this study were ordered from Sangon Biotech. DNA oligonucleotides were suspended in Milli-Q water (Millipore) at a concentration of 100 μM and stored at −20 °C prior to use. Buffer conditions were described as follows: Tris-HCl buffer with 20 mM Tris-HCl (pH 7.5, 22 °C), 100 mM NaCl, 0.1 mM EDTA, and 5 mM MgCl$_2$ for the reactions except for DNAzyme. For the DNAzyme experiment, the buffer conditions were Tris-HCl buffer with 20 mM Tris-HCl (pH 7.5, 22 °C), 100 mM NaCl, 0.1 mM EDTA, and 20 mM MgCl$_2$.

**Preparation of DNA samples**. All the preformed duplexes involved in this study were annealed in a T100 Thermal Cycler PCR (Bio-Rad). The samples were first heated up to 95 °C for 5 min and then slowly cooled to 20 °C at the constant rate of 1 °C/min. For annealing of all the reporter duplexes, the strand with the quencher group was in a ratio of 1.4 against the fluorophore-labeled strand to ensure the displacement signals could be clearly reported. After annealing, all the preformed duplexes were stored at 4 °C prior to use. Other DNA strands were directly diluted from the stock solution into the reaction mixture without any further treatment.

**Kinetic experiments of the ligand-induced fluorescence change**. The fluorescence was recorded by a SHIMADZU-RF-6000 fluorescence spectrophotometer using LabSolutions RF 1.11. For the reporter system with the FAM fluorophore, excitation/emission was at 480 nm/520 nm. For the reporter with the TMR

fluorophore, excitation/emission was at 550 nm/590 nm. Both excitation and emission bandwidths were 10 nm, and fluorescence data were recorded every 5 s for all the experiments. All the experiments were performed at room temperature (~22 °C). In a typical kinetic experiment of Figs. 2 and 3, all the DNA components including the reporter system were mixed together in the solution at room temperature. The concentrations of DNA components were 50 nM aptamer-containing strands and 25 nM reporter systems. The kinetic measurement was started upon the addition of the ligand. Background fluorescence measurements were carried out following the same procedure except that the addition of the ligand was replaced by the addition of the blank buffer. DNA Sequences utilized in these experiments were listed in Supplementary Tables 1–6. All the fluorescence data were presented and analyzed by GraphPad Prism 7.0.

**Kinetic experiments for the logic operations**. The OR experiments were carried out in the reaction buffers containing the Apt-Thr-OR strand (50 nM), the Apt-ATP-OR strand (50 nM), and the Rep-2 system (25 nM). Kinetic measurements were started upon the addition of designated concentrations of ATP or thrombin. For the NOT experiment, reaction buffers containing the Apt-Thr-NOT strand (50 nM), the Apt-ATP-NOT strand (200 nM), and the Rep-2 system (25 nM) with or without the designated ATP concentrations were incubated at room temperature for 5 min. Kinetic measurements were started upon the addition of the thrombin protein (200 nM). For the AND experiment, reaction buffers containing the Apt-Thr-AND strand (50 nM) with different lengths of toeholds, the Apt-ATP-AND strand (50 nM), and the Rep-2 system (25 nM) were mixed at room temperature for 5 min. Kinetic measurements were started upon the addition of designated concentrations of ATP or/and thrombin. Background fluorescence measurements were carried out following the same procedure except that the addition of the ligand was replaced by the addition of the blank buffer. DNA Sequences utilized in these experiments were listed in Supplementary Tables 7 and 8.

**Kinetic experiments for the cascade reactions**. Reaction buffers containing the Apt-Thr-A/B strand (50 nM), the preformed Apt-ATP-A/B duplex (annealed by 50 nM Apt-ATP-A/B and 60 nM Masking strand A/B) were incubated at room temperature for 15 min. Kinetic measurements were started upon the addition of designated concentrations of ATP or/and thrombin. Background fluorescence measurements were carried out following the same procedure except that the addition of the ligand was replaced by the addition of the blank buffer. For the experiments with stimulating or inhibiting feedback, reaction buffers containing the Apt-Thr-stimulation or Apt-Thr-inhibition strand (60 nM), the preformed triplex assembly annealed by 50 nM Apt-ATP, 50 nM Output-1, 50 nM Complementary-1 (Strand i), the preformed Output-2 duplex annealed by 50 nM Output-2 and 50 nM Complementary-2 (Strand ii) and the designated concentrations of ATP and thrombin were incubated at room temperature for 3 hr to achieve an efficient conversion. As a control for the system baseline, all the DNA components maintained the same except the Apt-Thr strand was removed. Fluorescence intensities were recorded after incubation with the Rep-4 duplex (100 nM) for another 0.5 h at the room temperature for the signal reporting process and the fluorescence increase was determined compared with the baseline control. DNA Sequences utilized in these experiments were listed in Supplementary Tables 9–11.

**Measurement of fluorescence change with signal amplifications**. For the experiments coupled with toehold exchange, reaction buffers containing the ligand transduction platform (50 nM Apt-ATP-TE or Apt-Thr-TE strand, and the 25 nM preformed duplex consisting of Apt-Output-TE and Apt-Complementary-TE strands), the toehold exchange system (500 nM TE-Fuel strand iii, and the preformed duplex consisting of 200 nM TE-Displacement strand i and 200 nM TE-Complementary strand ii) and the Rep-3 duplex (200 nM) were incubated at the room temperature for 30 min before addition of designated concentrations of ATP or thrombin. As a control for the system baseline, all the DNA components maintained the same except the Apt-ATP-TE or Apt-Thr-TE strand was removed. Fluorescence intensities were recorded after incubation for different time points at the room temperature and the fluorescence increase was determined compared with the baseline control. For the experiments coupled with DNAzyme, reaction buffers containing the ligand transduction platform (50 nM Apt-ATP-DNAzyme strand or 30 nM Apt-Thr-DNAzyme strand, and the preformed 25 nM duplex consisting of Apt-Output-DNAzyme and Apt-Complementary-DNAzyme strands for ATP or 12.5 nM duplex consisting of Apt-Output-DNAzyme and Apt-Complementary-DNAzyme strands for thrombin) and 200 nM DNAzyme-Substrate strand were incubated at the room temperature for 30 min before addition of designated concentrations of ATP or thrombin. This dual labeled DNAzyme-Substrate strand can be cleaved by the released DNAzyme strand (the Apt-Output-DNAzyme strand) in the presence of magnesium. As a control for the system baseline, all the DNA components maintained the same except the Apt-ATP-DNAzyme or Apt-Thr-DNAzyme strand was removed. Fluorescence intensities were recorded after incubation for different time points at the room temperature and the fluorescence increase was determined compared with the baseline

control. DNA Sequences utilized in these experiments were listed in Supplementary Tables 12 and 13. Reporter duplexes were listed in Supplementary Table 14.

**Reporting summary**. Further information on research design is available in the Nature Research Reporting Summary linked to this article.

## Data availability

The data that support the findings of this study are available from the corresponding author upon reasonable request. Source data are provided with this paper.

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

## Acknowledgements

The authors acknowledge funding from the National Natural Science Foundation of China (Nos. 21977122 and 21708054) and the Innovative Research Team in the University of Ministry of Education of China (IRT_17R111) and the National Key R&D Program of China (2020YFA0211200).

## Author contributions

L.X. designed the project. Q.-L.Z. carried out the experiments with assistance from L.-L.W., Y.L., and J.L. Q.-L.Z. and L.X. analyzed the data. L.X. supervised this work and wrote the paper.

## Competing interests

The authors declare no competing interests.
