## [Peer Review File · Nature Communications]

Reviewers' Comments:

Reviewer #1:

Remarks to the Author:

Qiu-Long Zhang et al. presented a new type of ligand-aptamer binding design. In their design, the aptamer domain is placed in the middle of the DNA strand; a toehold and a displacement domain are placed on both ends. Especially, they have a duplex stem between the aptamer domain and the toehold/displacement domains. The duplex stem can facilitate the subsequent strand-displacement reaction once the aptamer captures the target. They demonstrated their design first by an ATP aptamer and optimized the sequence of the duplex stem. They also show that their design strategy works very well on thrombin and PDGF protein induced strand-displacement. Besides, they designed the logical operations, including "OR," "NOT," and "AND," and cascade reactions by using the ATP and thrombin aptamers. Moreover, they show that their ligand-oligonucleotide transduction can be integrated with DNA toehold exchange-based catalysis and magnesium-based DNAzyme system.

The method is well presented and is demonstrated by reliable experiments and data.

Several Concerns:

1. Could you do experiments to determine the detection limitation of ATP, thrombin, and PDGF? This could show the potential in design biosensors by using your method.
2. Also, is it possible to detect ATP or thrombin in body fluids, such as serum?
3. It is necessary to label the DNA strands in the figures, such as Figure 6, so that readers can figure out the reaction flow easily.
4. Line 139, "the output strand can be fully random and independent of the aptamer sequence". This statement is too strong. Still need to avoid possible strong and stable secondary structures formed by the aptamer sequence and the toehold/displacement domains. Also, the reaction rate of subsequent strand-displacement needs to be taken into consideration.

Reviewer #2:

Remarks to the Author:

Summary

In this work, the authors present a strategy for ligand-oligonucleotide transduction that is both kinetically controlled and allows for release of an independent oligonucleotide. A known aptamer sequence is modified by the addition of short (~1-4 bp), complementary stem sequences on each end as well as a toehold domain beyond the stem regions on the 5' end and a displacement domain beyond the stem on the 3' end. Target binding induces a conformational change that allows duplex formation between the added stem sequences. The newly formed stem sequence then brings the added toehold and displacement domains together and allows them to act as an invading strand in toehold-mediated strand displacement. Kinetic control is afforded by optimization of the stem region duplex length and the length of the toehold region. Independence of the transduced oligonucleotide arises from the ability to select displacement region sequences completely independent of binding domains.

The proposed strategy was demonstrated for ATP sensing, mycotoxin ochratoxin A (OTA), thrombin, and platelet-derived growth factor, illustrating its compatibility with both small molecules and proteins. Additionally, the reported transduction system was also used to construct "OR", "NOT", and "AND" logical operation gates between ATP and thrombin ligands, demonstrate a system with a cascade relationship between different ligands, and couple ligand-oligonucleotide transduction other nucleic acid-based systems including catalytic systems for signal amplification and a DNAzyme cleavage.

This work presents a novel ligand-oligonucleotide transduction strategy that is potentially a meaningful contribution to the field of DNA nanotechnology and synthetic biology. However, the manuscript is not publishable in its current form, as it suffers from a number of critical shortcomings as described below. Thus, I suggest a major revision.

Major issues:

- First and foremost, the quality of writing is poor with many grammatical and syntax errors throughout the manuscript. The authors are strongly suggested to perform a thorough overhaul of the entire manuscript.

- Secondly, the authors are sloppy about presenting their results. The quality of presentation does not meet scientific standards. Many plots are missing error bars and triplicate data. The kinetic curves should be performed in triplicate as there may be experiment-to-experiment variation.

- Along similar lines, the authors are very qualitative in their description of their results. For example, on lines 187-189, the authors state that they chose a 3-bp stem because it had "negligible background leakage"... What was this background signal? How was it determined to be "negligible"? What was the signal to noise ratio of the constructs? Similarly, line 193, authors state "the kinetic difference was still clearly observed compared with the absence of ATP." This statement has no scientific or statistical meaning. They need to report an LOD or perform a statistical test of the signal vs. the background and report a p-value. Overall, the quality of presentation does not meet scientific standards for publication.

- Generally, this technique does not appear to provide enhancement in sensitivity over other standard aptamer-based techniques. A majority of the ligand-responsive functions are demonstrated at ligand concentrations far higher than the base aptamer affinities (eg. in the ATP case it is ~100-fold above K_d). The most promising data is presented in the amplification section (Figure 6) where 1 μ M ATP detection is shown, although this does require a long incubation of 4 hours. I would suggest the authors demonstrate this amplification technique with the other 3 ligand systems. If they could demonstrate enhanced sensitivity within a rapid measurement time that compares to other analytical techniques (e.g. ELISA), this impact of this work would be significantly enhanced.

- The value of the logic operations demonstrated in the manuscript is unclear. To achieve the same functionality, it seems entirely possible to implement these logic operations (OR, AND, NOT, cascade) downstream of the aptamer recognition using the output strands. The ability to perform these logic operations using simple released DNA strands is very common, thus the authors should provide a clear explanation of why it is more valuable to implement this directly with the aptamer-containing strand.

- In the first example, the authors use the knowledge about the secondary structure of the ATP aptamer to design their construct. i.e. it is known that the two ends of the ATP aptamer are brought close together upon binding to ATP. The authors state that no structural knowledge of the aptamer is needed, a claim that they make because they demonstrated that their approach also works with the OTA aptamer, of which they did not have secondary structure information. However, the entire method relies on the two ends of the aptamer being brought sufficiently close together upon target binding, which is not necessarily true of all aptamers. Therefore, there is not sufficient evidence to claim that this approach is generalizable to all aptamers.

- The authors state multiple times that "there are no sequence constraints" for the DNA output sequences. However, there are glaring constraints: obviously the invading strand can't be partially complementary to the aptamer or have any strong secondary structures.

- The authors must use the word "kinetics" more carefully. Sometimes it is used to mean the rate of signal generation, but it is also frequently used to describe the mechanism of action of the sensor—where signal is generated or suppressed based on the different rate constants of strand displacement for a both vs. unbound aptamer.

- Ligand concentration dependent kinetic curves should be measured and presented for OTA and PDGF.

- There does not appear to be a clear difference between the AND and cascade logic operations. Both systems are operated in the same way by simultaneously adding ATP and Thrombin, and both produce very similar kinetic curves. If the author could present some difference in function between the two systems, the cascade result would be more interesting.

- The NOT logic gate appears to depend on sequential events since prior ATP addition is needed to inhibit the Thrombin aptamer before thrombin is added. This function appears closer to a cascade measurement. The authors should demonstrate that this construct functions as intended for simultaneous ATP & Thrombin addition, or they need to be clearer in explaining the exact type of logic operation being performed.

- The authors claim on line 206 that they identified the “core binding sequence” of the OTA aptamer. It implies that they minimized the OTA aptamer to find its core binding domain – this is unclear. Similarly, in lines 166-169, there is no reference to where the ATP sequence came from. It appears to be a shortened version of the traditional ATP aptamer, but it is unclear where that came from.

Minor Issues:

- In line 106, the authors state that their approach overcomes the limitations of other “competition-based” approaches. However, it could be argued that their approach is also competition-based.
- The sentence on line 149 needs additional explanation
- Figure S2 should be fit to a Langmuir binding equation, not a linear line. Also, consider plotting the x-axis on a log scale.
- AND/OR gates with aptamers are not new. The article should cite at least one previous implementation, such as “Photonic Boolean logic gates based on DNA aptamers”, Chemical Communications 2007

Point-by-Point Response

Reviewer #1 (Remarks to the Author):

“Qiu-Long Zhang et al. presented a new type of ligand-aptamer binding design. In their design, the aptamer domain is placed in the middle of the DNA strand; a toehold and a displacement domain are placed on both ends. Especially, they have a duplex stem between the aptamer domain and the toehold/displacement domains. The duplex stem can facilitate the subsequent strand-displacement reaction once the aptamer captures the target. They demonstrated their design first by an ATP aptamer and optimized the sequence of the duplex stem. They also show that their design strategy works very well on thrombin and PDGF protein induced strand-displacement. Besides, they designed the logical operations, including “OR,” “NOT,” and “AND,” and cascade reactions by using the ATP and thrombin aptamers. Moreover, they show that their ligand-oligonucleotide transduction can be integrated with DNA toehold exchange-based catalysis and magnesium-based DNAzyme system.

The method is well presented and is demonstrated by reliable experiments and data.”

Response: We highly appreciate the reviewer’s positive comments.

“Several Concerns:

1. Could you do experiments to determine the detection limitation of ATP, thrombin, and PDGF? This could show the potential in design biosensors by using your method.”

Response: We highly appreciate the reviewer’s suggestion. In the revised manuscript, we have determined the limit of detection (LOD) of all the ligands investigated by our new platform. For instance, with the fluorescence-based measurement, the LOD of ATP was 7.6 μM without any signal amplification process, which could be further enhanced to reach sub-micromolar upon the signal amplification with either the toehold exchange system ($\sim 0.93 \mu\text{M}$) or the DNAzyme system ($\sim 0.75 \mu\text{M}$). The sensitivity of our method was fully comparable to the commercial ATP kits (typically $\sim 1 \mu\text{M}$). We believe future designs with employment of different signal sensing and amplifying techniques would greatly expand the applicable scenarios of our platform.

“2. Also, is it possible to detect ATP or thrombin in body fluids, such as serum?”

Response: We highly appreciate the reviewer’s suggestion. In the revised manuscript, we have included the recovery test (new **Figure S3** for ATP and **Figure S10** for thrombin) to check the performance of our method in the serum samples. These data suggested our platform can still effectively detect ATP and thrombin in the complex system.

“3. It is necessary to label the DNA strands in the figures, such as Figure 6, so that readers can figure out the reaction flow easily.”

Response: We thank the reviewer for the suggestion. The DNA strands involved in the signal amplification process have been labeled in the revised **Figure 6**.

“4. Line 139, “the output strand can be fully random and independent of the aptamer sequence”. This statement is too strong. Still need to avoid possible strong and stable secondary structures formed by the aptamer sequence and the toehold/displacement domains. Also, the reaction rate of subsequent strand-displacement needs to be taken into consideration.”

Response: We fully agree with this point and highly appreciate the reviewer for pointing out this inaccuracy in our description. In our revised manuscript, we have revised the expressions accordingly. We have also indicated the sequence requirement and discussed necessary considerations when designing this transduction system in the section of “designing principle” and the discussion part.

Reviewer #2 (Remarks to the Author):

Summary

“In this work, the authors present a strategy for ligand-oligonucleotide transduction that is both kinetically controlled and allows for release of an independent oligonucleotide. A known aptamer sequence is modified by the addition of short (~1-4 bp), complementary stem sequences on each end as well as a toehold domain beyond the stem regions on the 5' end and a displacement domain beyond the stem on the 3' end. Target binding induces a conformational change that allows duplex formation between the added stem sequences. The newly formed stem sequence then brings the added toehold and displacement domains together and allows them to act as an invading strand in toehold-mediated strand displacement. Kinetic control is afforded by optimization of the stem region duplex length and the length of the toehold region. Independence of the transduced oligonucleotide arises from the ability to select displacement region sequences completely independent of binding domains.

The proposed strategy was demonstrated for ATP sensing, mycotoxin ochratoxin A (OTA), thrombin, and platelet-derived growth factor, illustrating its compatibility with both small molecules and proteins. Additionally, the reported transduction system was also used to construct “OR”, “NOT”, and “AND” logical operation gates between ATP and thrombin ligands, demonstrate a system with a cascade relationship between different ligands, and couple ligand-oligonucleotide transduction other nucleic acid-based systems including

catalytic systems for signal amplification and a DNAzyme cleavage.

This work presents a novel ligand-oligonucleotide transduction strategy that is potentially a meaningful contribution to the field of DNA nanotechnology and synthetic biology. However, the manuscript is not publishable in its current form, as it suffers from a number of critical shortcomings as described below. Thus, I suggest a major revision.”

Response: We highly appreciate the reviewer’s positive comments as well as constructive critics. This manuscript has been carefully revised accordingly, and details can be found in the following response.

Major issues:

“• First and foremost, the quality of writing is poor with many grammatical and syntax errors throughout the manuscript. The authors are strongly suggested to perform a thorough overhaul of the entire manuscript.”

Response: We appreciate the reviewer’s critics for our language issues. In our revised manuscript, we have polished the manuscript thoroughly.

“• Secondly, the authors are sloppy about presenting their results. The quality of presentation does not meet scientific standards. Many plots are missing error bars and triplicate data. The kinetic curves should be performed in triplicate as there may be experiment-to-experiment variation.”

Response: We appreciate the reviewer for pointing out the issues in our data presentations. In this revised manuscript, we have improved the quality of data presentation. For qualitative description of kinetic behaviors of the transduction platform (such as sequence optimizations and logical operations), representative kinetic curves were shown in the data presentations. To quantitatively describe the detection limit of ligands, data with standard deviations from three independent replicates were presented in the revised manuscript (note that error bars shorter than the height of the symbol are not shown in the graph).

“• Along similar lines, the authors are very qualitative in their description of their results. For example, on lines 187-189, the authors state that they chose a 3-bp stem because it had “negligible background leakage”... What was this background signal? How was it determined to be “negligible”? What was the signal to noise ratio of the constructs? Similarly, line 193, authors state “the kinetic difference was still clearly observed compared with the absence of ATP.” This statement has no scientific or statistical meaning. They need to report an LOD or perform a statistical test of the signal vs. the background and report a p-value. Overall, the quality of presentation does not meet scientific standards for publication.”

Response: We highly appreciate the reviewer for pointing out the lack of statistics during our

data presentation. In the revised manuscript, statistical tests with P -values have been included into the quantitative description of our data. Moreover, LOD values have been also determined for all the ligands investigated in this work. These revisions can be found in the **Figure 2**, **Figure 3**, and **Figure 6** in the main text, and numerous new supplementary figures in the supporting data. In addition, the expressions in the data presentation have been also carefully revised accordingly.

“• Generally, this technique does not appear to provide enhancement in sensitivity over other standard aptamer-based techniques. A majority of the ligand-responsive functions are demonstrated at ligand concentrations far higher than the base aptamer affinities (eg. in the ATP case it is ~100-fold above K_d). The most promising data is presented in the amplification section (Figure 6) where 1 μ M ATP detection is shown, although this does require a long incubation of 4 hours. I would suggest the authors demonstrate this amplification technique with the other 3 ligand systems. If they could demonstrate enhanced sensitivity within a rapid measurement time that compares to other analytical techniques (e.g. ELISA), this impact of this work would be significantly enhanced.”

Response: We appreciate the reviewer’s point. In this manuscript, we reported both the basic design of this transduction platform and the signal amplification-coupled system. In fact, with the basic design without any signal amplification process, the LOD of ATP could reach 7.6 μ M and the LOD of thrombin was 0.96 nM, which was in the similar level of the K_d value of ATP (~6 μ M reported in *Biochemistry* 1995, 34, 656-665) or thrombin (~0.5 nM reported in *J. Mol. Biol.* 1997, 272, 688-698). Due to the designing flexibility and expandability, our platform could be easily coupled with signal amplification techniques to further enhance the detection limit. In this revised manuscript, in addition to the ATP measurement with our amplification systems, we have also included the detection of thrombin as a representative for the protein ligands to check the performance of our design (new **Figure S19-22**), and demonstrated that transduction of both small molecules and proteins could be coupled with the signal amplification systems based on our strategy. Notably, how fast the detection can proceed mainly depends on the amplification techniques. Herein, we selected the toehold exchange system to amplify the signal, which was previously reported to require several hours for the process (e.g. *J. Am. Chem. Soc.* 2009, 131, 17303-17314); however, in the DNAzyme system, the amplification was significantly faster with clear distinction observed within one hour (**Figure 6b** and **Figure S17**). It would be foreseeable that other kinds of nucleic acid-based signal amplification systems, including non-enzymatic catalysis as well as enzymatic amplifications (*Acc. Chem. Res.* 2014, 47, 1825-1835; *Chem. Rev.* 2019, 119, 6326-6369), could potentially provide more options for different requirements of analyte detections. We have included these points in our revised manuscript.

“• The value of the logic operations demonstrated in the manuscript is unclear. To achieve the same functionality, it seems entirely possible to implement these logic operations (OR, AND, NOT, cascade) downstream of the aptamer recognition using the output strands. The ability to perform these logic operations using simple released DNA strands is very common, thus the authors should provide a clear explanation of why it is more valuable to implement this directly with the aptamer-containing strand.”

Response: We highly appreciate the reviewer’s point. In the discussion part, we also mentioned that the logical operations could be achieved through the manipulation of the output strands. However, compared with the downstream operations, direct manipulation of the invading strands possesses one important advantage: ligand-induced interactions between aptamer-containing strands can be directly detected by the reporter system for signal transduction without any intermediate strand components. This reduced requirement for the strand components might not be necessarily critical for analytical purpose, but could be highly important for constructing genetic circuits in synthetic biology, as the simpler the system is in the cellular context, the more efficiency the signal transduction could be. We have included these explanations in the revised manuscript.

“• In the first example, the authors use the knowledge about the secondary structure of the ATP aptamer to design their construct. i.e. it is known that the two ends of the ATP aptamer are brought close together upon binding to ATP. The authors state that no structural knowledge of the aptamer is needed, a claim that they make because they demonstrated that their approach also works with the OTA aptamer, of which they did not have secondary structure information. However, the entire method relies on the two ends of the aptamer being brought sufficiently close together upon target binding, which is not necessarily true of all aptamers. Therefore, there is not sufficient evidence to claim that this approach is generalizable to all aptamers.”

Response: We highly appreciate the reviewer’s point regarding the possible structural requirement of aptamer in our system. Herein, we would like to make some explanations from three perspectives. First, we believe the restrained conformational dynamics of the aptamer strand, rather than the distance proximity of its two ends, is the key factor in our design. Generally, ligand binding would restrain the conformational dynamics of the aptamer structure, which could facilitate the stability of the linked short duplex, and *vice versa*. From this perspective, the dynamically conformational freedom of aptamer in the absence of ligand binding would reduce the thermostability of duplex formation; once the conformation of the aptamer is defined by ligand binding, the thermostability of the short duplex stem would be enhanced to boost the following strand displacement. Second, for the aptamers without any structure information, a couple of flanking nucleotides between the aptamer binding domain

and the short duplex stem were introduced to allow the free space for the aptamer folding and minimize possible impacts on the aptamer binding caused by the formation of duplex stem. Third, to the best of our knowledge, a well-characterized aptamer structure with two ends sterically far away from each other is not available from the literature. Herein, in addition to the OTA aptamer, we have selected another aptamer sequence without secondary structure information, the L-tyrosinamide (Tym) aptamer (*Anal. Chem.* 2016, 88, 11963–11971), to further verify the applicability of our design. As shown in new **Figure S6** and **S7**, the Tym ligand can also effectively control the strand displacement with introduction of an extra short duplex into the aptamer sequence. This additional example further supported the successful utility of our platform in the absence of aptamer structure information. We have included these explanations and new data in the revised manuscript.

“• The authors state multiple times that “there are no sequence constraints” for the DNA output sequences. However, there are glaring constraints: obviously the invading strand can’t be partially complementary to the aptamer or have any strong secondary structures.”

Response: We highly appreciate the reviewer for pointing out this inaccuracy in our description. As we have responded to Point 4 of Reviewer 1, we have revised the expressions and made explanations accordingly.

“• The authors must use the word “kinetics” more carefully. Sometimes it is used to mean the rate of signal generation, but it is also frequently used to describe the mechanism of action of the sensor—where signal is generated or suppressed based on the different rate constants of strand displacement for a both vs. unbound aptamer.”

Response: We highly appreciate the reviewer for pointing out this ambiguity. We have revised some words with the adjective form and utilized the word “rate” instead in some places.

“• Ligand concentration dependent kinetic curves should be measured and presented for OTA and PDGF.”

Response: We highly appreciate the reviewer’s point. In this revised manuscript, we have performed the concentration-dependent kinetic curves for OTA, Tym, and PDGF, and determined the LOD values for all these ligands as described in new **Figure S5, S7** and **S12**.

“• There does not appear to be a clear difference between the AND and cascade logic operations. Both systems are operated in the same way by simultaneously adding ATP and Thrombin, and both produce very similar kinetic curves. If the author could present some difference in function between the two systems, the cascade result would be more interesting.”

Response: We highly appreciate the reviewer’s suggestion. Different from the AND operation, the cascade process can provide a feedback signal against the initial input ligand. In our revised manuscript, we have presented two new feedback circuits based on the cascade design as described in new **Figure S14** and **S15**. As shown in **Figure S14**, the addition of ATP would stimulate the signal of thrombin, suggesting a positive feedback between these two ligands; in contrast, ATP could inhibit the thrombin signal in the other design (**Figure S15**) due to the negative feedback effect. These new feedback operations explicitly demonstrated the important function of the cascade design based on this transduction platform. Detailed data and description can be found in the revised manuscript.

“• *The NOT logic gate appears to depend on sequential events since prior ATP addition is needed to inhibit the Thrombin aptamer before thrombin is added. This function appears closer to a cascade measurement. The authors should demonstrate that this construct functions as intended for simultaneous ATP & Thrombin addition, or they need to be clearer in explaining the exact type of logic operation being performed.*”

Response: We highly appreciate the reviewer’s point. In the NOT gate, we allowed the interactions between the ATP and thrombin aptamer-containing strands to occur before the reporting process. Alternatively, we have also performed another experiment with a different procedure, in which the ATP and thrombin were pre-mixed together with the aptamer-containing strands before the kinetic measurement of the reporting process. As shown in **Figure R1**, these kinetic data (Procedure 2) were similar as the previous procedure (Procedure 1). In either case, interactions between the aptamer-containing strands were needed before the reporting process as the inhibition and the strand displacement was in a direct competition. We have included these explanations in the revised manuscript.

Figure R1. Comparison between two different experimental procedures for the NOT gate. In Procedure 1, ATP and all the strands including the reporter duplex pre-existed into the system

to inhibit the thrombin aptamer before addition of thrombin; In Procedure 2, ATP, thrombin and two aptamer-containing strands were mixed together before addition of the reporter duplex. Thrombin: 200 nM; ATP: 1 mM.

“• The authors claim on line 206 that they identified the “core binding sequence” of the OTA aptamer. It implies that they minimized the OTA aptamer to find its core binding domain – this is unclear. Similarly, in lines 166-169, there is no reference to where the ATP sequence came from. It appears to be a shortened version of the traditional ATP aptamer, but it is unclear where that came from.”

Response: We highly appreciate the reviewer’s point. In the revised manuscript, we have cited the sources of all these aptamer sequences used in this work. For the OTA aptamer, different truncations have been reported in the literature (*Agr. Food Chem.* 2008, 56, 10456-10461; *Anal. Bioanal. Chem.* 2013, 405, 2443–2449). Herein, we utilized the consensus sequence of this aptamer, which have been described in the revised manuscript.

Minor Issues:

“• In line 106, the authors state that their approach overcomes the limitations of other “competition-based” approaches. However, it could be argued that their approach is also competition-based.”

Response: We understand the reviewer’s point. Our expression of the “competition-based” approach indicated the competitive binding with the aptamer between the target ligand and the masking strand. We have made more explanations in the revised manuscript.

“• The sentence on line 149 needs additional explanation”

Response: We highly appreciate the reviewer’s point. Since the conformational dynamics of aptamer might be the key to govern the strand displacement, a larger distinction of the strand dynamics between the loose and the defined conformation would provide more space to manipulate the kinetic rates of strand displacement. A short sequence of DNA strand is, typically, less loose than a long strand in terms of the conformational perspective. If a small dynamic change of a short strand can be clearly distinguished, a longer sequence of strand with more dramatically conformational variation would be more applicable for manipulation. We have included these explanations in this revised manuscript.

“• Figure S2 should be fit to a Langmuir binding equation, not a linear line. Also, consider plotting the x-axis on a log scale.”

Response: We thank the reviewer’s suggestions. With analysis of all the investigated ligands,

we found that the linear lines were more suitable to fit the values with low ligand concentrations (new **Figure S2, S5, S7, S9** and **S12**), and the LOD values could be easily determined by the canonical 3σ /slope method. In the systems with the signal amplification process (toehold exchange catalysis and DNAzyme), we found that utilization of a log scale for x-axis would be appropriate to fit the values (new **Figure S18, S21** and **S22**). All these fitting processes have been described in the revised manuscript.

“• AND/OR gates with aptamers are not new. The article should cite at least one previous implementation, such as “Photonic Boolean logic gates based on DNA aptamers”, Chemical Communications 2007”

Response: We thank the reviewer for mentioning this reference. We have included this citation in the revised manuscript.

Reviewers' Comments:

Reviewer #1:

Remarks to the Author:

All my previous concerns have been well addressed.

Only one question:

Did you use 1% serum samples in the recovery test for detection of thrombin? In the table of Figure S10, it says 10% serum.

Reviewer #2:

Remarks to the Author:

In this resubmission, the authors present a novel ligand-oligonucleotide transduction strategy that offers a simple and kinetically tunable way to leverage aptamer-based sensing and DNA strand displacement reactions with great control and flexibility. This work represents a meaningful advancement in the fields of DNA nanotechnology and biosensing, worthy of publication. However, the manuscript still does not meet the standards for publication, in terms of quality of writing. For example, there are numerous errors in diction ('verity' rather than 'variety' on line 43), improper use of an adverb in place of an adjective ('precisely kinetic control' rather than 'precise kinetic control' on line 14) and use of informal language where scientific language would be more appropriate ('less loose' rather than 'more entropically confined' or something similar in line 149). I therefore recommend the paper for publication, after it goes through a thorough editing process.

Reviewer #1 (Remarks to the Author):

“All my previous concerns have been well addressed.

Only one question:

Did you use 1% serum samples in the recovery test for detection of thrombin? In the table of Figure S10, it says 10% serum.”

Response: It was 1% serum in the table of Figure S10. This error has been corrected in the revised manuscript.

Reviewer #2 (Remarks to the Author):

“In this resubmission, the authors present a novel ligand-oligonucleotide transduction strategy that offers a simple and kinetically tunable way to leverage aptamer-based sensing and DNA strand displacement reactions with great control and flexibility. This work represents a meaningful advancement in the fields of DNA nanotechnology and biosensing, worthy of publication. However, the manuscript still does not meet the standards for publication, in terms of quality of writing. For example, there are numerous errors in diction (‘verity’ rather than ‘variety’ on line 43), improper use of an adverb in place of an adjective (‘precisely kinetic control’ rather than ‘precise kinetic control’ on line 14) and use of informal language where scientific language would be more appropriate (‘less loose’ rather than ‘more entropically confined’ or something similar in line 149). I therefore recommend the paper for publication, after it goes through a thorough editing process.”

Response: We have carefully polished and edited our manuscript thoroughly. These language issues have been revised.